# Development of Apoptotic-Cell-Inspired Antibody–Drug Conjugate for Effective Immune Modulation

**DOI:** 10.3390/ijms242216036

**Published:** 2023-11-07

**Authors:** Gyeongwoo Lee, Taishu Iwase, Shunsuke Matsumoto, Ahmed Nabil, Mitsuhiro Ebara

**Affiliations:** 1Research Center for Macromolecules and Biomaterials, National Institute for Materials Science (NIMS), 1-1 Namiki, Tsukuba 305-0044, Japantolba.ahmednabil@nims.go.jp (A.N.); 2Graduate School of Pure and Applied Sciences, University of Tsukuba, 1-1-1 Tennodai, Tsukuba 305-8577, Japan; 3Biotechnology and Life Sciences Department, Faculty of Postgraduate Studies for Advanced Sciences (PSAS), Beni-Suef University, Beni-Suef 62511, Egypt; 4Graduate School of Advanced Engineering, Tokyo University of Science, 6-3-1 Katsushika-ku, Shinjuku, Tokyo 125-8585, Japan

**Keywords:** immune modulation, antibody–drug conjugates, antibody–polymer conjugates, bio-inspired polymer

## Abstract

Background: Apoptotic cells’ phosphoserine (PS) groups have a significant immunosuppressive effect. They inhibit proinflammatory signals by interacting with various immune cells, including macrophages, dendritic cells, and CD4^+^ cells. Previously, we synthesized PS-group-immobilized polymers and verified their immunomodulatory effects. Despite its confirmed immunomodulatory potential, the PS group has not been considered as a payload for antibody–drug conjugates (ADCs) in a targeted anti-inflammatory approach. Aim: We conducted this research to introduce an apoptotic-cell-inspired antibody–drug conjugate for effective immunomodulation. Method: Poly(2-hydroxyethyl methacrylate-*co*-2-methacryloyloxyethyl phosphorylserine) (p(HEMA-*co*-MPS)) was synthesized as a payload using RAFT polymerization, and goat anti-mouse IgG was selected as a model antibody, which was conjugated with the synthesized p(HEMA-*co*-MPS) via 1-Ethyl-3-(3-dimethylaminopropyl)carbodiimide/*N*-Hydroxysuccinimide (EDC/NHS) reaction. The antibody-binding affinity, anti-inflammatory potential, and cytotoxicity measurements were evaluated. Results: We successfully synthesized ADCs with a significant anti-inflammatory effect and optimized the antibody–polymer ratio to achieve the highest antibody-binding affinity. Conclusion: We successfully introduced p(HEMA-*co*-MPS) to IgG without decreasing the anti-inflammatory potential of the polymer while maintaining its targeting ability. We suggest that the antibody–polymer ratio be appropriately adjusted for effective therapy. In the future, this technology can be applied to therapeutic antibodies, such as Tocilizumab or Abatacept.

## 1. Introduction

Antibody–drug conjugates (ADCs) have shown high therapeutic efficiency and diminished cytotoxicity compared with conventional treatment [1]. Currently, 14 ADCs are approved by the FDA: gemtuzumab/ozogamicin, brentuximab/bedotin, adotrastuzumab/emtansine, inotuzumab/ozogamicin, polatuzumab/vedotin, enfortumab/vedotin, fam-trastuzumab/deruxtecan, Ioncastuximab/tesirine, disitamab/vedotin, tisotumab/vedotin, sacituzumab/govitecan, belantamab/mafodotin, cetuximab/sarotalocan, and moxetumomab/pasudotox. All ADCs were approved for oncological diseases; however, to date, no ADCs have been approved for non-oncological diseases. Although ADCs are mostly used in cancer therapies, their applications are expanding to non-cancer issues [2]. Many researchers have attempted to apply ADCs to various diseases, including muscular diseases, infections, nephritis, rheumatoid arthritis, and atherosclerosis [3,4,5,6,7]. Several ADCs are currently being tested in clinical trials for non-oncological diseases. ABBV-3773 is a combination of adalimumab (an anti-tumor necrosis factor) and a glucocorticoid receptor modulator for rheumatoid arthritis. This was tested in a phase 2 study and showed an enhanced therapeutic effect compared to that of conventional adalimumab. However, serious adverse events were twice more frequently reported in ABBV-3773-treated patients compared to adalimumab-treated patients [8]. DSTA4637S is an anti-*Staphylococcus aureus*–rifamycin conjugate that has no severe adverse effects on healthy volunteers; however, approximately 25% of patients showed infusion-related reactions [9,10]. These results indicate that safety is important for successful ADC development. 

Although antibody conjugation greatly enhances the biological effects of the payload, an effective and safe payload should be selected for successful ADC development [11]. In doxorubicin–antibody conjugates, the tumor-inhibitory effect of doxorubicin was greatly enhanced in an animal model [12]. However, the antibody–doxorubicin conjugate did not show a significant therapeutic effect in a phase 2 study [13]. In contrast, dexamethasone, a synthetic glucocorticoid, was conjugated with antibodies and showed improved therapeutic effects in an animal model without any adverse effects [14]. Other glucocorticoid-receptor-modulator-conjugated ADCs show increased adverse effects compared to the non-drug-conjugated antibody in phase 2 studies; nonetheless, the conjugated drug has fewer adverse effects than dexamethasone [8]. These results indicate that the payload should be carefully selected, considering both the effectiveness and safety for the successful development of ADCs.

Apoptotic cells are natural immune suppressors in the body. They play crucial roles in the resolution of inflammation and immune modulation [15]. Macrophages interact with various immune cells, and most apoptotic cells interact with macrophages via their phosphoserine (PS) groups. These groups interact with macrophages and polarize the macrophage phenotype from classically stimulated M1 to alternatively stimulated M2 macrophages. After polarization, macrophages stop secreting inflammatory cytokines and contribute to the resolution of inflammation through the clearance of dead cells and the secretion of anti-inflammatory cytokines [16]. Their interactions with dendritic cells suppress the secretion of proinflammatory cytokines, in addition to dendritic cell maturation inhibition [17]. Recently, the suppression of CD4^+^ cells, owing to their interactions with apoptotic cells in PS groups, has also been reported. By injecting apoptotic cells into a mutated mouse, the T-cell signaling pathway and rheumatoid arthritis are suppressed [18]. Despite the excellent therapeutic effects of apoptotic cells, adverse effects have not yet been reported.

To the best of our knowledge, the mechanism of action of apoptotic cells is not fully understood. Nonetheless, the PS group is an important factor in the therapeutic effects of apoptotic cells. Macrophages exhibit similar immunosuppressive reactions when treated with PS-group-containing liposomes. However, when treated with liposomes lacking PS, no suppressive effects are observed [19]. A similar tendency was observed in dendritic cells treated with liposomes that do not contain PS, and dendritic cell maturation is not suppressed [20]. In addition, CD4^+^ cell activation is inhibited after recognition of the PS group [18]. This evidence suggests that PS groups are strongly associated with their therapeutic effects. 

We previously synthesized an apoptotic-cell-inspired 2-methacryloyloxyethyl phosphoserine (MPS) polymer and verified its anti-inflammatory effects [21]. Furthermore, we fabricated PS-group-containing polymer nanoparticles and verified their therapeutic effects in lipopolysaccharide (LPS)-injected mice [22]. PS-group-immobilized polymers are considered to have equivalent therapeutic effects on apoptotic cells. Therefore, our study demonstrates the great potential of PS-group-immobilized polymers as ADC payloads. Despite their great potential as payloads for safe and effective ADC therapy, PS-group-immobilized polymers have not been studied as ADC payloads. In this study, we designed, synthesized, and characterized an antibody–polymer conjugate of goat anti-mouse IgG as a model antibody and poly(2-hydroxyethyl methacrylate (HEMA)-*co*-MPS). Goat anti-mouse IgG and p(HEMA-*co*-MPS) were conjugated by the 1-Ethyl-3-(3-dimethylaminopropyl)carbodiimide/*N*-Hydroxysuccinimide (EDC/NHS) reaction at different feed ratios, and the binding affinity and immunosuppressive effects were evaluated to establish the optimized polymer introduction ratio as an ADC payload, as shown in (Figure 1).

## 2. Results and Discussion

### 2.1. Synthesis of p(HEMA-co-MPS)

The (*t*-BuO/Boc) MPS monomer was synthesized via a phosphoramidite reaction and copolymerized with 2-hydroxyethyl methacrylate (HEMA) via radical polymerization (Figure 1). Polymerization was conducted using free-radical and reversible addition–fragmentation chain-transfer polymerization (RAFT) polymerizations, and the molecular weight distributions were compared (Appendix A, Table 1). The free-radical-polymerized copolymer showed M_n_ and M_w_ of 194 kg/mol and 410 kg/mol, respectively. On the other hand, the RAFT-polymerized copolymer showed M_n_ and M_w_ of 9.5 kg/mol and 13.5 kg/mol, respectively. The polydispersity indexes were 2.1 and 1.4, respectively. The RAFT-polymerized copolymer exhibited a narrow molecular weight distribution. Hence, the RAFT-polymerized copolymer was used in subsequent experiments.

The successful synthesis of p(HEMA-*co*-MPS) was confirmed by ^1^H NMR spectroscopy (Appendix A). In the spectra of the (*t*-BuO/Boc)MPS monomer, the characteristic peaks of the HEMA monomer were observed, including diene (5.7 and 6.0 ppm), methyl (1.85 ppm), and ethyl (3.7–4.2 ppm). In addition, the characteristic peak of the protective group (*t*-BuO/Boc, 1.3 ppm) was observed (Appendix A). After the copolymerization of (*t*-BuO/Boc)MPS and HEMA, the diene peak disappeared, whereas methyl, ethyl, and protective group peaks were observed (Appendix A). This result indicated the successful copolymerization of p(HEMA-*co*-(*t*-BuO/Boc)MPS). After deprotection, other characteristic peaks were observed; however, the characteristic peaks of the protective groups were significantly diminished (Appendix A). This result suggests successful deprotection, which agrees with our previous report [21]. The characteristic peaks of the Boc and *t*-Bu groups completely disappear under acidic conditions. The introduction of the PS group was verified using a fluorescamine assay (Appendix A). p(HEMA-*co*-(*t*-BuO/Boc)MPS) did not show any fluorescence intensity. In contrast, the deprotected p(HEMA-*co*-MPS) showed a significant increase in fluorescence intensity. These findings suggest that the PS group was successfully introduced without any damage to the molecule during deprotection. 

### 2.2. Conjugation of Goat Anti-Mouse IgG with p(HEMA-co-MPS)

Goat anti-mouse IgG and p(HEMA-*co*-MPS) were conjugated using EDC/NHS coupling (Figure 1). The carboxylic acid in p(HEMA-*co*-MPS) was activated by EDC in the presence of NHS and reacted with the amine group of goat anti-mouse IgG. The molecular weights of goat anti-mouse IgG and IgG-p(HEMA-*co*-MPS) were determined by sodium dodecyl sulfate–polyacrylamide gel electrophoresis (SDS-PAGE) (Figure 2). Goat anti-mouse IgG showed bands at 50 kDa and 25 kDa. We conclude that goat anti-mouse IgG was cleaved to the heavy and light chains by 2-mercaptoethanol. In contrast, IgG-p(HEMA-*co*-MPS) showed a broad molecular weight band at approximately 150 kDa, regardless of the feed ratio between the antibody and the polymer. These findings suggest that p(HEMA-*co*-MPS) was successfully introduced to goat anti-mouse IgG. In our previous report, significant band broadening was observed after the conjugation, and these previous observations are consistent with the current results [23]. However, bands at around 50 and 25 kDa were not observed for IgG-p(HEMA-*co*-MPS), unlike in our previous report [23]. We conclude that a single chain of p(HEMA-*co*-MPS) reacted with several amine groups on IgG, and a bridge was formed between the heavy and light chains instead of a disulfide bond in IgG (Figure 2a). Hence, the heavy- and light-chain bands were not observed. In addition, all IgG-p(HEMA-*co*-MPS) showed bands over 250 kDa. More than two antibodies were connected using p(HEMA-*co*-MPS). To observe the molecular weight profile, GPC was measured for both the model IgG and IgG-p(HEMA-*co*-MPS) 1:2 (Appendix A). Although the elution curves indicate a 9% increase in molecular weight after conjugation, curve separation was not observed. This indicates that the molar ratio of the bridged antibody is not high enough to be observed at this polymer feed ratio.

No increase in molecular weight was observed with an increase in the antibody–polymer feed ratio. In one of our previous reports, a broad band was observed after the conjugation of the polymer, and the median spot shifted to a high molecular weight with an increase in the antibody–polymer feed ratio [24]. Our findings are inconsistent with these previous results, as the reactive moiety of the antibody was saturated at a polymer feed ratio of 1:5.

Unreacted p(HEMA-*co*-MPS) was observed near the loading well of the gel before the gel was fully washed (Appendix A) because p(HEMA-*co*-MPS) exhibited a white band below 25 kDa. In contrast, IgG-p(HEMA-*co*-MPS) 1:2 did not produce any white bands. IgG-p(HEMA-*co*-MPS) 1:5 showed a weak white band, the intensity of which increased with increasing antibody–polymer feed ratios. These results indicate that p(HEMA-*co*-MPS) completely reacted with goat anti-mouse IgG at a ratio of 1:2, and the reaction was fully saturated in IgG-p(HEMA-*co*-MPS) at a ratio of 1:5 (Figure 2b). Considering these data, the antibody–polymer ratio of IgG-p(HEMA-*co*-MPS) 1:2 is estimated to be 2. On the other hand, IgG-p(HEMA-*co*-MPS) 1:5 has a ratio of less than 5. Since all IgG-p(HEMA*-co-*MPS) showed similar molecular weight bands in the SDS-PAGE results, it is inferred that all IgG-p(HEMA*-co-*MPS) conjugates have a similar antibody–polymer ratio close to 2. To determine a therapeutic effect, it is important to ensure the appropriate conjugation content of the drug. Although high drug concentrations show high biological activity, excessive conjugation can lead to the denaturation of antibodies and the loss of antibody specificity [25]. Therefore, both the antibody-binding affinity and therapeutic effects should be considered when optimizing the polymer conjugation content.

### 2.3. Antibody-Binding Affinity after Conjugation with p(HEMA-co-MPS)

The antibody-binding affinity was determined using sandwich ELISA (Figure 3, Appendix A.). Based on our baseline, which is the 100% binding affinity of goat anti-mouse IgG, the antibody-binding affinity linearly decreased with increasing polymer feed ratios. IgG-p(HMEA-*co*-MPS) 1:2 showed approximately 92% antibody-binding affinity, whereas IgG-p(HEMA-*co*-MPS) 1:5 showed approximately 88% antibody-binding affinity. IgG-p(HEMA-*co*-MPS) 1:10 showed significantly decreased antibody-binding affinity, with a reduction of approximately 75%. This result agrees with a previous study conducted by Yant et al. [26]. Because conjugation can occur throughout the antibody, a high antibody–polymer conjugation ratio induces stoichiometric hindrance and the denaturation of the antibody, and the antibody-binding affinity is highly attenuated [25]. Until the antibody–polymer feed ratio reached 1:5, the antibody-binding affinity did not significantly decrease. In contrast, an antibody–polymer feed ratio of 1:10 showed significantly decreased binding affinity. We conclude that an excessive polymer feed ratio affected the antibody variable region and attenuated its binding affinity.

### 2.4. Cell Viability and Anti-Inflammatory Effect of p(HEMA-co-MPS)

The cytotoxicity of p(HEMA-*co*-MPS) in macrophages was evaluated using the AlarmarBlue assay (Figure 4a). Macrophages showed excellent cell viability (approximately 100%) in the concentration range from 10 mg/mL to 10 μg/mL. Our previous study showed that the MPS polymers were not cytotoxic [21]. In the present study, we copolymerized MPS and poly-HEMA to fabricate p(HEMA-*co*-MPS). Because poly-HEMA is a non-toxic polymer widely used in the biomedical field, p(HEMA-*co*-MPS) did not show any cytotoxicity [27]. Macrophages showed similar cell viability regardless of lipopolysaccharide (LPS) stimulation. Macrophages undergo programmed cell death in response to the co-stimulation of toll-like receptors and scavenger receptors [28]. Our results indicated that neither the HEMA nor MPS repeating units stimulated scavenger receptors. This suggests that p(HEMA-*co*-MPS) suppressed inflammation without macrophage depletion.

The anti-inflammatory effect of p(HEMA-*co*-MPS) was evaluated by quantifying IL-6 secretion by ELISA (Figure 4b). IL-6 secretion was approximately 500 pg/mL without p(HEMA-*co*-MPS) treatment and decreased in a concentration-dependent manner. At a polymer concentration of 10 mg/mL, IL-6 secretion decreased to approximately 50 pg/mL. This finding is consistent with those of our previous report [21]. The anti-inflammatory effect of PS was maintained regardless of HEMA copolymerization. In addition, IL-6 secretion was approximately 10 times suppressed without any adverse effects on cell viability.

### 2.5. Anti-Inflammatory Effect of IgG-p(HEMA-co-MPS)

The IgG-p(HEMA-*co*-MPS) anti-inflammatory effect was increased after antibody–polymer conjugation, as confirmed by the IL-6 ELISA (Figure 5). IgG-p(HEMA-*co*-MPS) 1:2 and 1:5 showed greater suppression of IL-6 secretion than p(HEMA-*co*-MPS). This result is in agreement with Thomsen et al.’s previous study [14]. Antibody–drug conjugation has been reported to improve drug activity. In the present study, conjugation with goat anti-mouse IgG improved the anti-inflammatory effects of p(HEMA-*co*-MPS); however, the antibody used was goat anti-mouse IgG. The Fc region of IgG is believed to enhance the effect of p(HEMA-*co*-MPS). IgG binds to Fc receptors in macrophages, regardless of the target [29]. Therefore, conjugation with IgG improved the accessibility of the PS group and its receptor, leading to an enhanced effect. In addition, M1 macrophages have upregulated expression levels of Fc receptors [30]. Therefore, conjugation with IgG is expected to improve the targetability of inflammatory lesions because of both the variable and Fc regions. Moreover, increasing the polymer feed ratio inhibited the polymer’s anti-inflammatory potential, and this was observed clearly with the upregulation of IL-6 secretion by macrophages. According to the SDS-PAGE results, polymer conjugation was saturated at a 1:5 feed ratio. Therefore, it is inferred that an excessive polymer concentration led to the structural denaturation of the polymer and antibody. Furthermore, excessive combination with p(HEMA-*co*-MPS) showed a diminished antibody-binding affinity in our study. Therefore, the antibody–polymer ratio should be appropriately adjusted for effective therapy.

The relative gene expression of proinflammatory and anti-inflammatory markers was determined using qPCR (Table 2). Both p(HEMA-*co*-MPS) and IgG-p(HEMA-*co*-MPS) showed the downregulation of the proinflammatory markers IL-6, TNF-α, and iNOS. In contrast, the anti-inflammatory markers CD206 and IL-10 were upregulated. TGF-β1 expression was decreased. To the best of our knowledge, macrophage gene expression in response to stimulation with PS-group-conjugated polymers has not been previously reported. However, our results are similar to those for apoptotic cells. Apoptotic cells downregulate proinflammatory cytokines and upregulate anti-inflammatory cytokines [19]. In the present study, p(HEMA-*co*-MPS) had a similar effect to that of apoptotic cells. Therefore, p(HEMA-*co*-MPS) is considered to share the same signaling pathway with apoptotic cells. Apoptotic cells show anti-inflammatory effects through the activation of the AKT 1/2/3 pathway and blocking of the NF-κB and p38-MAPK signaling pathway [31]. Hence, if other anti-inflammatory antibodies with different mechanisms of action are conjugated with p(HEMA-*co*-MPS), a synergistic effect between the antibody and p(HEMA-*co*-MPS) is expected for inflammatory disease treatment.

Generally, the antibody–drug conjugation dosage ranges from 1 to 2 mg/kg [32]. The concentration of p(HEMA-*co*-MPS) used in this study was 10 mg/mL. Although the concentrations could not be compared directly, they were at least 100 times higher than the general dosage. However, in a previous study, the anti-inflammatory drug dexamethasone was tested at a concentration range of 1–10^−7^ mg/mL. The authors conducted an animal study that included dexamethasone administration with a 1 mg/kg dosage. The dexamethasone–antibody conjugate was 50-fold more effective than the compared dexamethasone monotherapy [33]. Therefore, it is expected that p(HEMA-*co*-MPS) will also show significant therapeutic effects not only in vitro but also at the animal level. If the therapeutic effects of p(HEMA-*co*-MPS) are verified in animals, p(HEMA-*co*-MPS) will have great potential as a payload in the ADC field.

## 3. Materials and Methods

### 3.1. Materials

HEMA, dichloromethane (DCM), imidazole hydrochloride, 2,2′-azobis(isobutyronitrile) (AIBN), and 2-(*N*-morpholino)ethanesulfonic acid (MES) were purchased from Wako Pure Chemical Industries (Osaka, Japan). 1-Ethyl-3-(3-dimethylaminopropyl)carbodiimide (EDC) and trifluoroacetic acid (TFA) were purchased from Tokyo Chemical Industry (Tokyo, Japan). *N*-Boc-*L*-serine *tert*-butyl ester was purchased from Chem-Impex International (Wood Dale, IL, USA). *N*-Hydroxysuccinimide (NHS), *tert*-butyl tetraisopropyl, and phosphorodiamidite were purchased from Sigma (St. Louis, MO, USA). Mouse IgG (Isotype control, Ab37355), goat anti-mouse IgG (ab6708), and HRP-conjugated goat mouse IgG (Ab6789) were purchased from Abcam (Boston, MA, USA). IL-6 uncoated ELISA kit, RAW-Blue cells, AlamarBlue_,_ and qPCR primers were purchased from Invitrogen (Carlsbad, CA, USA). The RNeasy kit was purchased from Qiagen (Venlo, The Netherlands). iScript Reverse Transcription Supermix for RT-qPCR and iTaq Universal SYBR green Supermix were purchased from BIO-RAD (Berkely, CA, USA).

### 3.2. Synthesis of MPS Monomer

The MPS monomer was synthesized as we previously described [21]. First, HEMA was distilled to remove stabilizers. Then, 5 g of *N*-Boc-*L*-serine *tert-*butyl ester (24.36 mmol), 5.243 g of *tert*-butyl tetraisoprypylphosphordiamidite (17.22 mmol), and 0.574 g of imidazole hydrochloride were added to 129 mL of DCM and stirred for 21 h at 25 °C under a nitrogen atmosphere. Afterward, 2.205 mL of HEMA (18.18 mmol) and 1.88 g of imidazole hydrochloride were added, and the mixture was stirred. After 45 min and 90 min, 1.88 g of imidazole hydrochloride was added repeatedly with stirring. After 150 min, the solution was washed with Milli-Q water, and the separated DCM phase was collected and dehydrated overnight with sodium sulfate. Subsequently, the DCM was evaporated, and the monomer was separated by column chromatography. The synthesis of the MPS monomer was verified by ^1^H NMR spectroscopy (JEOL, Tokyo, Japan). The details of ^1^H NMR are available in the Appendix A.

### 3.3. Synthesis of p(HEMA-co-MPS)

p(HEMA-*co*-MPS) was prepared via free-radical polymerization or RAFT polymerization. Briefly, 0.0943 mL of HEMA (0.78 mmol), 360 mg of MPS monomer (0.78 mmol), and 0.254 mg of AIBN were added to 3 mL of DMF. For RAFT polymerization, 20 mg of CDSPA (0.05 mmol) was added. Afterward, the solution was stirred for 21 h at 40 °C under a nitrogen atmosphere. After the reaction, the solution was dialyzed for 2 days in DCM. After dialysis, a partial aliquot was dried to measure molecular weight through gel permeation chromatography, and TFA was added to 25% (*v*/*v*) of the solution and stirred for 4 h to deprotect the *t*-butyl and Boc groups. The details of gel permeation chromatography are available in the Appendix A. Subsequently, all solvents were evaporated, and the remaining product was dissolved in Milli-Q water. After dialysis in 0.01 M NaOH aqueous for 1 day and Milli-Q water for another day, p(HEMA-*co*-MPS) was obtained by lyophilization. Deprotection of the PS group was observed via a fluorescamine assay. The details of fluorescamine are available in the Appendix A.

### 3.4. Fabrication of IgG-p(HEMA-co-MPS)

Firstly, MES buffer was prepared by dissolving 9.6 g of MES in 450 mL of Milli-Q water, and the pH was adjusted to 6.02 with 1 M NaOH. Goat anti-mouse IgG was prepared at a concentration of 1.33 × 10^−5^ M. The prepared goat anti-mouse IgG and MES buffer were added to a 3 k centrifugation tube with a 300 µL volume. Goat anti-mouse IgG was washed three times with MES buffer by centrifugation at 13,400 rpm for 5 min. After washing, goat anti-mouse IgG was resuspended in 300 µL of MES buffer. At the same time, p(HEMA-*co*-MPS) was prepared with various concentrations using MES buffer (2.66 × 10^−5^ M, 6.65 × 10^−5^ M, 1.33 × 10^−4^ M, and 6.65 × 10^−4^ M). A volume of 100 µL of IgG solution, 100 µL of p(HEMA-*co*-MPS) solution, 6 mg of EDC, and 6 mg of NHS were mixed in a tube rotator for 6 h. Next, glycine (6 mg) was added and mixed using a tube rotator for 15 min. The reacted IgG was washed three times with PBS by centrifugation in a 50 k centrifugation tube. Conjugation of IgG-p(HEMA*-co-*MPS) was verified via GPC and SDS-PAGE. Details are available in the Appendix A.

### 3.5. Measurement of the Binding Affinity

A sandwich enzyme-linked immunosorbent assay (ELISA) was used to measure the apparent binding affinity of APCs using the method in our previous report [34]. Goat anti-mouse IgG, a primary antibody, was dissolved in a coating buffer (pH 9.6, 1.0 mg/mL), stabilized in a 96-well cell culture plate, and incubated overnight. Following stabilization, the plate was washed three times with PBS containing 0.5% Tween 20 as a washing buffer. Next, the plate was blocked with 200 µL of blocking buffer and incubated for 30 min. The plate was then washed three times with PBS containing 0.5% Tween 20. A volume of 100 µL of mouse IgG as a model antigen was added to the 96-well plate with an antigen concentration of 1 ng/mL and incubated for 1 h. The plates were then washed three times with the washing buffer. Thereafter, 2 μg/mL HRP-conjugated IgG or IgG-p(HEMA-*co*-MPS) was added to 100 μL. After another three washes, 100 µL of TMB was added. After 15 min, 100 µL of 1 M hydrochloric acid was added. The absorbance of each well was measured at 450 nm by using a plate reader (Infinite 200 PRO, Tecan, Swiss).

### 3.6. Cytotoxicity and Anti-Inflammatory Potential Evaluation

To evaluate both the cytotoxicity and anti-inflammation potential, RAW-Blue macrophage (Sigma-Aldrich, St. Louis, MO, USA) cells were purchased and seeded into a 96-well plate at a density of 5 × 10^4^ cells/well and cultured in Dulbecco’s modified Eagle’s medium (DMEM; Nacalai, San Diego, CA, USA) supplemented with 10% fetal bovine serum (FBS; Sigma-Aldrich, St. Louis, MO, USA) and 1% penicillin–streptomycin (Sigma-Aldrich, USA) in a CO_2_ incubator. After 24 h, 10 μL of 4 μg/mL LPS in PBS was added to each well so that macrophages were polarized to M1 macrophages. After 30 min, each well was aspirated and washed with PBS once, and 150 μL of DMEM was added again. After that, 30 μL of each sample was added to each well, including PBS, p(HEMA-*co*-MPS) (2.66 × 10^−6^ M), and IgG-p(HEMA-*co*-MPS) (1.33 × 10^−6^ M). After 24 h, the cell supernatant was collected, and AlamarBlue solution was added to each well according to the manufacturer’s instructions to evaluate cell viability. After 3 h, absorbance at 570 nm was measured using a plate reader (Infinite 200 PRO, Tecan, Mannedorf, Switzerland). IL-6 secretion was quantified using an ELISA kit following the manufacturer’s instructions.

### 3.7. Statistical Analysis

Data are expressed as means ± standard deviations unless otherwise stated. Statistical comparisons were performed by one-way ANOVA with Tukey’s multiple-comparison test using Sigma Plotsoftware (Version 13, Systat Software, Sanjose, CA, USA). Statistical significance was set at *p* < 0.05.

## 4. Conclusions

In this study, an apoptotic-cell-inspired polymer (p(HEMA-*co*-MPS)) was synthesized via RAFT polymerization, followed by goat anti-mouse IgG conjugation via an EDC/NHS reaction. Different polymer feed ratios of 2, 5, 10, and 50 were reacted with the same IgG concentration, and based on our SDS-PAGE investigation, the unreacted polymer was observed after a 1:5 antibody–polymer feed ratio. Interestingly, the conjugated antibody showed the optimal binding affinity compared to the native antibody until a 1:5 ratio. The highest anti-inflammatory potential was observed at the 1:5 ratio without any cytotoxicity. These results indicate that the antibody–polymer ratio should be appropriately adjusted to achieve the optimal combined effect. In the future, further animal experiments should be conducted to verify the anti-inflammatory effects of p(HEMA-*co*-MPS) conjugated with therapeutic antibodies, including Tocilizumab or Abatacept.

## Data Availability

All data generated or analyzed during this study are included in this article. The raw data are available from the corresponding author upon reasonable request.

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
