# Peer review of "Development of Apoptotic-Cell-Inspired Antibody–Drug Conjugate for Effective Immune Modulation"

_ijms, 2023, doi:10.3390/ijms242216036_

Round 1

Reviewer 1 Report

Comments and Suggestions for Authors

Lee et al. report on the synthesis of an antibody-polymer conjugate of goat anti-mouse IgG (as a model antibody) and a p(HEMA-co-MPS) copolymer (poly(2-hydroxyethyl methacrylate-co-2-methacryloyloxyethyl phosphorylserine), showing anti-inflammatory effects). The copolymer was synthesized via RAFT polymerization and then conjugated to IgG via EDC/HMS reaction at different polymer feeding ratios. Antibody binding affinity as well as cytotoxicity and anti-inflammatory potential were evaluated indicating that the antibody/polymer ratio should be appropriately adjusted to achieve the optimal combined effect with regard to effectiveness and safety.

My impression is of a well written manuscript that could be of interest to the readership of IJMS. The subject addressed is worthy of investigation; the information presented is new, and conclusions are supported by data. I recommend accepting of the manuscript. Some minor points can be considered before accepting of the manuscript:

-        Scheme 1 is not mentioned in the text. Introduce, mention, refer to, discuss Scheme 1;

-        The values of Mn and Mw in g/mol (lines 95, 96 and Table 1) are non-sense because they are too small. Perhaps, kg/mol or 10^3 were meant;

Describe the methods of GPC, NMR, Fluorescence, and SDS-PAGE electrophoresis in the Supplementary Materials.

Comments on the Quality of English Language

Minor editing of English language is required.

Reviewer 2 Report

Comments and Suggestions for Authors

“Development of Apoptotic Cell Inspired Antibody-Drug Conjugate Toward Effective Immune Modulation” by Gyeongwoo Lee, Taishu Iwase, Shunsuke Matsumoto, Ahmed Nabil, Mitsuhiro Ebara (Manuscript ID: ijms-2633748)

This manuscript described a new type of antibody-drug conjugates that contain an immunosuppressive apoptotic cells phosphoserine group (PS). The authors synthesized the PS payload and conjugated it onto random lysine residues of a goat anti-mouse antibody. The manuscript further characterized the resulting ADCs on antibody binding affinity, impact on cell viability, IL6 secretion of macrophages, and marker gene expression on macrophages.

The topic of the manuscript is interesting and therapeutically important. However the manuscript needs to address a number of issues prior to its acceptance for publication:

1)     The resulting ADCs described in the manuscript need more biophysical characterization such as size exclusion chromatography, especially when high molecular weight species are detected in Fig.2 (Page 5).

2)     Even though the normal concentration of 2-mercaptoethanol that can reduce heavy chain and light chain of  goat anti-mouse IgG cannot reduce the resulting ADCs, a concentration titration of reductants should be tested to see if the molecules could be reduced.   

3)     Although the resulting ADCs are known to be a mixture, it would be of interest to estimate the range of payload-antibody ratios for the molecular population.

4)     Table S1 is an important result and therefore should be included into the main text.

5)     In Fig.3 (Page 6), what is the antigen for this goat anti-mouse IgG used for the affinity binding assay? What is this goat anti-mouse IgG? There is no source information about this important antibody in Page 9 section 3.5.

6)     What are the RAW-Blue Macrophage cells used for Figure 5 (Page 8)? There is no source information about this cell line in Page 10 section 3.6.

7)     The figure legends for Fig.2, 3, 4, 5, and Table 1 & S1 (new table 2) are too concise and need more experimental description for figure explanation.

8)     In the introduction section of Page 1, please list the full names for all FDA-approved ADCs. In fact, all approved ADCs are for oncology indication only, even though some clinical-staged ADCs are for other indications. 

Round 2

Reviewer 2 Report

Comments and Suggestions for Authors

The revised manuscript is much improved and should be accepted for publication.